# Microbial Contributions to Heavy Metal Phytoremediation in Agricultural Soils: A Review

**DOI:** 10.3390/microorganisms12101945

**Published:** 2024-09-25

**Authors:** Zobia Khatoon, Ma. del Carmen Orozco-Mosqueda, Gustavo Santoyo

**Affiliations:** 1Institute of Chemical and Biological Research, Universidad Michoacana de San Nicolas de Hidalgo, Morelia 58030, Mexico; 2Departamento de Ingeniería Bioquímica y Ambiental, Tecnológico Nacional de México en Celaya, Celaya 38010, Mexico

**Keywords:** PGPB, microbial synergies, meta-organism, plant–microbe associations, endophytes

## Abstract

Phytoremediation is a sustainable technique that employs plants to reinforce polluted environments such as agroecosystems. In recent years, new strategies involving the plant microbiome as an adjuvant in remediation processes have been reported. By leveraging this microbial assistance to remediate soils contaminated with heavy metals such As, Pb, Cd, Hg, and Cr, plants can sequester, degrade, or stabilize contaminants more efficiently. Remarkably, some plant species are known for their hyper-accumulative traits in synergy with their microbial partners and can successfully mitigate heavy metal pollutants. This sustainable biotechnology based on plant–microbe associations not only aids in environmental cleanup but also enhances biodiversity, improves soil structure, and promotes plant growth and health, making it a promising solution for addressing agro-pollution challenges worldwide. The current review article emphasizes the potential of synergistic plant–microbe interactions in developing practical and sustainable solutions for heavy metal remediation in agricultural systems, which are essential for food security.

## 1. Introduction

The growing world population, accompanied by industrial development, mining, and urban sprawl, is accompanied by increased pollution of aquatic, aerial, and terrestrial ecosystems [1]. Soil pollution in agroecosystems is an intensifying global issue because it has significant implications for food security, environmental protection, and agricultural efficiency, depending on the human population [2]. Unsustainable farming practices and unnecessary agrochemical use have raised concerns regarding soil pollution since the Green Revolution [3]. Current statistical analysis has revealed that a substantial proportion of agronomic lands worldwide have been affected by heavy metal pollution, including cadmium (Cd), copper (Cu), chromium (Cr), mercury (Hg), lead (Pb), zinc (Zn), and nickel (Ni), mainly tainting arable soils [4]. According to Chen et al., metalloid arsenic (As) is frequently included in this category because of its analogous chemical properties and environmental behavior [5].

Soil pollution in agricultural areas is unevenly distributed globally, and some regions of the world are aggravated by this issue, mostly owing to their extensive industrialization. Research has shown that in countries such as China and India, a significant portion of the arable land is compromised by heavy metal pollution [6,7]. This problem not only affects crop yield and quality, but also poses profound hazards to the health of producers and consumers. For example, prolonged exposure to arsenic dust can have adverse effects on human health, together with the development of dermal lesions, peripheral neuropathy, and skin cancer. In addition, metals such as lead (Pb), mercury (Hg), and cadmium (Cd) can penetrate the human body through multiple routes, including the inhalation of dust or vapor, the consumption of contaminated food, or direct skin contact [8]. Upon entering the body, they can accumulate in tissues and organs, triggering a broad spectrum of adverse effects, such as neurological damage; cognitive impairment; kidney, liver, and cardiovascular injury; damage to DNA and protein levels; and increased risk of cancer [9]. Consequently, heavy metal exposure is a pressing public health issue that requires careful administration and preventive measures to protect human health [10].

Phytoremediation is a key environmental protection practice that uses soil plants to remediate harmful elements in the environment [10]. Phytoremediation is cost-effective, affordable, and less harmful to the environment in its immediate surroundings than other conventional practices [11]. Hyperaccumulator plant species are highly tolerant of heavy metals. Some examples include *Nymphaea tymphaea, Bornmuellera tymphaea, Pteris vittate, Sedum alfredii,* and *Thlaspi caerulescens* [1,2,6,11,12]; however, there are still challenges to overcome, such as slow growth rates and low biomass production [12]. Plant genome engineering can sometimes address this issue in plant species lacking these characteristics or with low heavy metal tolerance.

Microorganisms constantly surround plants and can create robust, long, and beneficial associations that increase their suitability for growth in adverse environments, such as soils contaminated by heavy metals [13]. Some bacteria (e.g., *Burkholderia* spp., *Pseudomonas* spp., *Bacillus* spp., *Rhizobium* spp., and *Frankia* spp.) and fungi (e.g., *Glomus mosseae* and *Trichoderma* spp.) [2] are also extremely tolerant to heavy metals so that they can endure in rhizospheric environments [14]. In the rhizosphere, the plant excretes root exudates, with nutrients utilized by the soil microbiota to grow and reproduce. Some phyla, such as Firmicutes, Proteobacteria, Bacteroidetes, and Acidobacteria, have highly tolerant species with resistance mechanisms to heavy metal toxicity and are common inhabitants of the rhizosphere [15]. Some of these bacterial groups also colonize and penetrate plant roots, launching themselves as endophytes. From the endosphere, they assist plants in increasing their tolerance and remediation of heavy metals in the soil [16]. Similarly, some fungi, including arbuscular mycorrhizal fungi (AMF), may stimulate phytoremediation through phytoextraction and phytostabilization. Thus, AMF can reduce the stress induced by heavy metals in their plant hosts and curb the processes of metal translocation within the plant [17].

In this study, we explored soil remediation mechanisms carried out by plants and the auxiliary role of their accompanying microbial partners in phytoremediation. We also focused on bacterial and fungal groups that synergize with plants to regain agricultural soils, which are indispensable for food safety and human well-being.

## 2. Strategies for Phytoremediation: The Current State-of-the-Art

Phytoremediation is a green and environmentally friendly practice that implements plants to eradicate detrimental elements from the environment, predominantly the soil, water, and air. This method affects the natural aptitudes of plants to absorb, gather, degrade, and assuage contaminants, making it a promising solution for addressing pollution in several ecosystems, including agronomic soils polluted with heavy metals [18]. Numerous phytoremediation technologies, such as phytoextraction, phytostabilization, phytodegradation, phytovolatilization, rhizofiltration, and rhizodegradation, are used to exterminate organic and inorganic chemicals from polluted places (each having a diverse means of action) that depend on the cleanup level, the pollutant type, plant types, and the state of the site [19]. Figure 1 shows the different assisted phytoremediation strategies used in this study.

Co-planting and plant growth-promoting rhizobacterial (PGPR) treatment have an encouraging effect on the phytoextraction process. Intercropping and endophytic plant growth-promoting bacterial inoculation (*Burkholderia phytofirmans* PsJN^T^) augmented the phytoextraction of Zn, Pb, and Cd directed to *Brassica juncea* “Małopolska” grown in a monoculture or co-planted with *Zea mays* L. “Codimon” and *Medicago sativa* L. “Sanditi” [20]. The association of *Bornmuellera nymphaea*—*Noccaea nymphaea* (NB) and *Bornmuellera tymphaea*–*Alyssum murale* (AB) inoculated with PGPR increased Ni in roots by 105.8 and 66.4%, respectively, in AB and NB covers, and 39.9 and 79.6% in the shoots. Hence, the collective application of the hyperaccumulator plants *N. nymphaea andB.tymphaea* administered with one of the two PGPR strains (strain NB24), derived from the rhizosphere of this mixed cover, seemed to be a thought-provoking option for efficient Ni phytoextraction to be verified in the field [21]. Metal accumulation (NiCl_2_, Pb(CH_3_COO)_2_, CuSO_4_, NaAsO_2_, and MnCl_2_) has been reported to be improved by *Miscanthus* × *giganteus* and rhizobacterial association along with plant growth in flotation tailings due to PGP properties (indole-3-acetic acid and siderophore production, 1-aminocyclopropane-1-carboxylic acid deaminase activity, and phosphate solubilization) [22].

The impact of phytostabilization in the management of HM-contaminated soils using *Koelreuteria paniculata* as a test plant considerably amended soil conditions, particularly by enhancing pH levels, fertility, and water retention. The treatment group exhibited a noticeable decline in the bioavailability and migration of heavy metals, such as Zn, Pb, and Mn, with a decrease in runoff loss of 15.7%, 8.4%, and 10.2%, respectively [23]. The potential of rice in connection with the PGPR consortium improved the comparative richness of bacteria related to di (2-ethylhexyl) phthalate degradation (*Sphingomonas*, Xanthobacteraceae), heavy metal immobilization (*Massilia*), and soil nutrient cycling (*Nitrospira,* Vicinamibacterales), which increased plant growth and heavy metal removal from soil, as well as Di (2-ethylhexyl) phthalate [24].

Phytovolatility has made a notable contribution to the detoxification of heavy metals. It remains a key mechanism in arsenic (As) decontamination during the bioremediation process involving *Arundo donax* L. and the amicrobial assistance of epigenetic changes by a PGPR consortium. PGPR consists of two strains of *Stenotrophomonas maltophilia* and one *Agrobacterium* sp. strain. The authors analyzed the methylation-sensitive amplified polymorphisms (MSAPs) of *A. donax* leaves and detected potential changes by interaction with As and the bacterial consortium. The presence of PGPB did not modify the percentage of volatilized arsenic or induce significant plant growth biomass, but the work demonstrated that phytovolatilization is a robust mechanism for reducing arsenic concentrations in contaminated environments [25].

The rhizofiltration proficiency of *Juncus acutus* L. was evaluated in a constructed wetland (CW) scheme that was certainly premeditated to treat groundwater polluted with hexavalent chromium (Cr (VI)). *J. acutus* efficiently eliminated up to 140 μg/L of Cr (VI) from contaminated groundwater. In addition, this study highlighted the role of endophytic bacteria associated with *J. acutus* in enhancing the plant’s resistance to Cr (VI) toxicity and its total filtration efficacy. *Pseudomonas* sp. strain R16 exhibited an extraordinary ability to reduce Cr (VI) to the less toxic trivalent form (Cr (III)) under aerophilic conditions. The decrease of 100 mg/L Cr(VI) by *Pseudomonas* sp. strain R16 after 150 h of incubation underscores the synergistic rapport between the plant and its endophytic bacteria in detoxifying Cr(VI) in contaminated environments [26].

In summary, the amalgamation of co-planting approaches, PGPR injection, and the practice of hyperaccumulators and resistant plant species has shown significant potential in enhancing such processes for the effective remediation of heavy metal-contaminated soils, highlighting an auspicious and synergistic method for environmental decontamination and sustainable land management.

## 3. Metal-Resistant Microbes

The isolation and application of microbial populations for the remediation of heavy metal ions from the environment have been a focus of particular interest owing to their capacity to effectively highlight pollution challenges. Microbial populations, predominantly those encompassing bacteria and fungi, have acquired various mechanisms to depollute and detain heavy metals, making them suitable candidates for bioremediation [27]. A list of bacterial species that support plants in soil remediation includes *Pseudomonas*, *Bacillus*, *Variovorax*, *Klebsiella*, *Sinorhizobium*, *Enterobacter*, *Rhodococcus*, *Flavobacterium*, *Ensifer*, and *Kocuria*, to mention but a few [1,2,10,12,13,19]. On the other hand, fungi from the genera *Trichoderma*, *Beauveria*, and various arbuscular mycorrhizal fungi (AMF) such as *Rhizophagus* and *Glomus* have established their contributions in serving their plant hosts in different soil-cleaning processes, with those that are part of agroecosystems [28].

### 3.1. PGPB Contributions

The explanation of plant-advantageous bacteria, largely metal-resistant strains, and the contemplation of their mechanisms to augment plant growth and metal tolerance is pivotal for undertaking phytoremediation-enhancing tactics. Table 1 presents the number of studies directed in this context using PGPR and PGPEs (plant growth-promoting endophytes). Plant health in metal-polluted soils is upgraded in two ways by these bacteria: first, through the direct production of plant growth-beneficial products together with the solubilization/transformation of mineral nutrients (e.g., phosphate, nitrogen, and potassium), the synthesis of phytohormones, siderophores, and definite enzymes; and second, they indirectly support plants by controlling plant pathogens or by bringing a systemic resistance of plants against pathogens. Furthermore, they alter the metal buildup volume in plants by eliminating metal-immobilizing extracellular polymeric substances and metal-mobilizing organic acids and biosurfactants [29].

In a recent study, ten bacterial strains from plant rhizospheres in mining deposits, recognized as *Enterobacter*, *Serratia*, *Klebsiella*, and *Escherichia* species, have shown significant potential for phytoremediation. Strain *Serratia* sp. K120 showed a positive correlation between ACC deaminase action and indole-3 acetic acid (IAA) production across numerous heavy metal treatments, including Pb, As, and Cu. This strain boosted plant growth under exposure to these metals and displayed resilience to the adverse effects of Ni, Cd, and Mn, making it a favorable candidate for use in phytoremediation systems aimed at remediating soils contaminated with heavy metals [30].

An inspection of *Pseudometallophytes* (*Betula celtiberica*, *Cytisus scoparius*, and *Festuca rubra*) in a Pb/Zn mine revealed the influence of metal-tolerant rhizobacteria on phytoremediation. Regardless of their low diversity, these bacteria, principally Actinobacteria, were more rampant in rhizosphere soils and presented tolerance to higher levels of Cd and Zn. Although not all isolates had PGP traits, many of them implicitly improved the growth of *Festuca pratensis* and *Salix viminalis*, signifying their potential for phytostabilization and phytoextraction efforts [31]. Eleven Cd-resistant bacterial strains, including *Variovorax paradoxus*, *Rhodococcus* sp. and *Flavobacterium* sp., were obtained from the root area of *Brassica juncea* planted in Cd-contaminated soils. These variants showed tolerance to various metals (Cd, Zn, Cu, Ni, and Co) and promoted root extension in *B. juncea* seedlings. Most strains, exclusive of *Flavobacterium* sp. strain 5P-3, formed ACC deaminase and were associated with enhanced root growth. These bacteria display potential as inoculants to advance *B. juncea* growth in Cd-polluted soils, supporting their use in phytoremediation strategies [32].

The diazotrophic rhizobacteria *Azospirillum baldaniorum* Sp245 (formerly *A. brasilense*) and *Azospirillum brasilense* Sp7 are two species capable of colonizing different habitats. The former (Sp245) is considered a facultative endophyte, as it can live either as an endophyte or saprophyte, while the Sp7 strain can only colonize the surface of roots. In a study by Kamnev and colleagues [33], they observed that these *Azospirillum* strains respond differently to the presence of heavy metals. The wild-type Sp245 strain showed a less marked buildup of stress-induced compounds compared to the non-endophytic type strain Sp7 when exposed to metals such as Co^2+^, Cu^2+^, and Zn^2+^. Moreover, both strains revealed reduced indole-3-acetic acid (IAA) production in the presence of Cu^2+^ or Cd^2+^, which hypothetically affects their efficiency as plant growth promoters in soils contaminated with heavy metals. This result shows that although there are naturally non-resistant strains to heavy metals such as *A. baldaniorum*, there are others that can tolerate broader ranges and could be candidates for inoculation in plants, thereby stimulating their fitness and growth, and in turn, indirectly improving remediation processes. The application of heavy metal (HM)-resistant PGPR variants *Ralstonia eutropha* (B1) and *Chrysiobacterium humi* (B2) sharply reduced Zn and Cd bioaccumulation in *Helianthus annuus*, with *C. humi* (B2) reducing Zn accumulation by up to 67%, Zn bioconcentration factor (BCF) by 20%, Zn uptake by 64%, and Cd uptake and BCF by 27%, whereas it enhanced bacterial diversity in the rhizosphere, thus improving plant stabilization in contaminated soils [34]. These findings emphasize the vital role of bacteria in the progress of phytoremediation efforts, predominantly in heavily polluted environments.

**Table 1 microorganisms-12-01945-t001:** Relevant studies demonstrating the role of plant growth-promoting rhizobacteria and endophytic bacteria in the phytoremediation of metal-contaminated soils, including those used for agriculture.

Type	Species/Strain	Plant Used	Mechanism of Action	Impact	References
Plant Growth-Promoting Rhizobacteria	*Sinorhizobium meliloti*	*Medicago lupulina* L.	IAA, ACC deaminase, and siderophores.	Amended biomass, elevated copper uptake, and decreased copper stress.	[35]
	*Variovorax paradoxus*	*Bornmuellera tymphaea* (Hausskn.) Hausskn. *Noccaea tymphaea* (Hausskn.) F.K.Mey. *Alyssum murale* L.	IAA, ACC deaminase, siderophores, and P solubilization.	Improved biomass and greater nickel uptake.	[21]
	*Chryseobacterium humi Pseudomonas reactans* *Pseudomonas fluorescens*	*Zea mays* L.	ACC deaminase, P solubilization, and siderophores.	Augmented root and shoot growth, higher biomass, and enhanced cadmium uptake.	[36]
	*Ensifer adhaerens*		Indole-3-acetic acid (IAA), siderophores, and ACC deaminase.	Heightened arsenic uptake.	[37]
	*Bacillus licheniformis* *Micrococcus luteus*	*Vitis vinifera* cv. *Malbec*	Nitrogen assimilation, P solubilization, and Siderophore.	Diminished toxicity from arsenic.	[38]
	*Kocuria* sp. CRB15	*Brassica nigra* L.	Indole-3-acetic acid (IAA) and solubilization of phosphorus.	Enhanced growth of roots and shoots.	[39]
	*Sinorhizobium Saheli*	*Leucaena leucocephala* (Lam.) de Wit	Fixation of nitrogen, solubilization of phosphorus, and synthesis of IAA.	Increased root and shoot growth, elevated biomass, decreased cadmium uptake, and reduced manganese uptake.	[40]
	*Pseudomonas* sp.	*Medicago sativa* L.	N fixation, P solubilization, IAA biosynthesis, and siderophore generation	Improved root and shoot development, greater biomass, boosted chlorophyll levels, decreased oxidative stress, and elevated Cr accumulation in roots.	[41]
	*Bacillus* sp. EhS7 *Acinetobacter* sp. RA1 *Bacillus* sp. RA2	Perennial ryegrass Tall fescue	IAA biosynthesis and P solubilization.	Greater biomass, less oxidative stress, and reduced uptake of metals.	[42]
Plant growth-promoting endophytic bacteria	*Bacillus* sp. E2S2 *Bacillus* sp. E1S2	*Sedum plumbizincicola* X.H.Guo and S.B.Zhou ex L.H.Wu	Indole-3-acetic acid (IAA) synthesis, ACC deaminase enzyme activity, phosphorus solubilization, and siderophore production.	Boosted root and shoot growth, augmented biomass, increased cadmium uptake, and higher zinc accumulation.	[37]
	*Serratia* sp. AI001 *Klebsiella* sp. AI002	*Solanum nigrum* L.	IAA production.	Increase in biomass, elevation of chlorophyll content, and enhancement of Cd translocation.	[43]

### 3.2. Fungal Interventions

During mycoremediation, the degradative power of fungi is used to eliminate or deactivate disparaging chemicals present in soil and water [44]. Dark septate endophyte (DSE), *Exophiala pisciphila,* triggered a noticeable tolerance to Cd, with a considerable decline in the toxic effects of Cd and a noteworthy increase in maize expansion by activating antioxidant systems, changing metal chemical forms into inactive Cd, and repartitioning subcellular Cd into the cell wall [45]. Exposure of *P. libanensis* autonomously or along with *C. claroideum* upgraded plant growth and reformed the physiological status (e.g., electrolyte leakage, chlorophyll, proline, and malondialdehyde contents) in addition to Ni and sodium (Na^+^) buildup potential (e.g., uptake and translocation factor of Ni and Na^+^) of *H. annuus* under Ni and salinity stress either alone or in combination [46]. The introduction of AMF, specifically *Funneliformis mosseae* (Fm) and *Rhizophagus intraradices* (Ri), to upland rice grown in soil containing 0, 2, or 10 mg Cd kg^−1^ not only reduced cadmium accumulation in rice but also tempered reactive oxygen species (ROS) scavenging activities [47]. The symbiont of *S. calendulacea* with FM provides a theoretical basis and application direction for the remediation of Cd-contaminated soil [48]. AMF were found to be very helpful in removing mercury from rice. Rice plants grown in symbiosis with AMF had much lower mercury levels, with a reduction of between 52.82% and 96.42% compared to rice plants without AMF [49].

A beneficial fungus called *Glomus versiforme* (Gv) facilitated the growth of upland rice in soil with added Cd. Rice plants with Gv grew taller, absorbed more phosphorus, and had healthier photosynthesis [50]. Planting a cadmium-absorbing plant, *Solanum nigrum*, in conjunction with rice in contaminated soil, with the help of beneficial fungi, reduced Cd in rice while encouraging plant growth [51].

*Trichoderma* is a genus of fungi central to agroecosystems because of its multifunctional properties that foster plant health and soil quality. These fungi are recognized for their ability to act as biocontrol agents against various plant pathogens, plummeting the need for chemical pesticides and promoting sustainable agricultural practices [52]. *Trichoderma* spp. boost plant growth by fabricating growth-promoting substances, solubilizing phosphates, and improving nutrient uptake, which results in increased crop yields. Additionally, *Trichoderma* subsidizes soil remediation by breaking down organic matter and decomposing soil pollutants, thereby improving soil structure and fertility. Their presence also promotes beneficial microbial communities, creating a more resilient and balanced soil ecosystem [53]. *Trichoderma* is essential in agroecosystems, as it provides environmental benefits and supports sustainable agriculture by enhancing plant growth and health.

*Trichoderma* can also support its plant hosts in cleansing agricultural soils, as verified in a study by Kumar and Dwivedi, who appraised the chromium-reducing potential of *T. lixii* strain CR700. In addition, strain CR700 can tolerate high concentrations of metals such as Cr (1000 mg/L), As (2000 mg/L), Ni (1500 mg/L), Zn (1200 mg/L), Cu (1200 mg/L), Pb (100 mg/L), and Cd (100 mg/L) [54]. Eventually, the authors proved that the phytotoxicity test using the supernatant from *T. lixii*-treated 100 mg/L Cr (VI) on *Vigna radiata* and *Cicer arietinum* showed effective detoxification and remediation of Cr (VI).

*Beauveria bassiana* is an important entomopathogenic fungus that has been extensively documented for its role in biological pest control. As naturally occurring pathogens of various insect species, it is a feasible substitute for chemical pesticides in managing agricultural pests, thus promoting ecologically sound farming practices. *B. bassiana* contaminates and kills insects through direct contact by infiltrating their cuticle, growing internally, and releasing toxins, ultimately leading to host death. Its broad host adaptability includes several major agricultural pests, such as aphids, beetles, and caterpillars, making it a multipurpose tool in integrated pest management (IPM) programs [55]. In addition, *B. bassiana* is environmentally friendly as it explicitly targets insects without harming beneficial organisms, humans, or the environment. Its capacity to persist in the environment and reprocess over successive generations of insects has further developed its utility as a long-term pest control plan. Overall, *Beauveria bassiana* is helpful in helping ecological balance and reducing reliance on synthetic pesticides in agricultural ecosystems [56].

Some cases showing the potential of *Beauveria* fungi include the work of Gola et al., who stated that a strain of *Beauveria bassiana* unveiled a high capacity for eliminating multiple metals from polluted wastewater, attaining an 84% removal rate, which is significantly higher than the 61–75% removal rate of individual metals [56]. The proclivity of the fungus to different metals also changed when exposed to a multimodal environment, representing its adaptability to complex adulteration scenarios. Atomic force microscopy (AFM) revealed changes in the surface roughness of *B. bassiana* due to metal noxiousness, suggesting alterations in its cell structure and function. These findings highlight the potential of *B. bassiana* as a promising strain for efficient multimodal metal removal in wastewater treatment. Even though this work is not directly linked to agroecosystems, this strain can be practical in water treatment for irrigation in agriculture, a topic that has been little explored in recent research but represents significant potential for future investigation.

In a recent evaluation, Kumar and Dwivedi studied the role of fungi in heavy metal removal and the factors influencing their effectiveness [28]. Thus, industrial development and coal and metal mining pollute water bodies with heavy metals, creating severe ecological hazards. The authors advocate that traditional treatment methods are costly and produce hazardous waste, making affordable and eco-friendly solutions essential. Therefore, bioremediation using fungi offers a sustainable approach, as fungi effectively adsorb and accumulate heavy metals through various mechanisms like bioaccumulation and bioabsorption, which were also discussed. Finally, the productivity of fungal bioremediation depends on ecological aspects such as time, pH, temperature, HM concentration, and fungal biomass. Studies on the involvement of fungi in phytoremediation induced by heavy metals are presented in Table 2.

**Table 2 microorganisms-12-01945-t002:** Recent works on the exploration of fungal roles in phytoremediation of heavy metals in different agrosystems.

Fungal Species	Host Plant	Heavy Metals	Consequences	Reference
*Rhizophagus irregularis*	*Cannabis sativa* L.	Cd remediation	Enhanced Cd compartmentation	[57]
*Laccaria*, *L. bicolor* and *L. japonica*	*Pinus densiflora* Siebold and Zucc.	Cadmium (Cd) or copper (Cu)	Blocked the migration and accumulation of cadmium	[58]
Ectomycorrhizal (ECM) fungi	*Pinus densiflora* Siebold and Zucc.	Cu	Increased seedling performance	[59]
*Paxillus involutus*	*P*. × *canescens*	Pb	Increased plant growth and may increase Pb phytostabilization potential	[60]
*Pisolithus albus*	*Acacia spirorbis* Labill. and *Eucalyptus globulus* Labill.	Co, Cr, Fe, Mn and Ni	Enhanced plant growth and mineral nutrition while limiting metal uptake	[61]
*Chaetomium globosum*	*Zea mays* L.	Copper	Increased seedling dry weight, osmotic solute content, and antioxidant enzyme activity	[62]
*Trichoderma harzianum Rifai 1295-22*	*Salix fragilis*	Cadmium manganese nickel and zinc	Promoted growth	[63]
*Glomus intraradices*	*Linum usitatissimum*	Nickel	Alleviated Ni toxicity as indicated by improved plant growth	[64]
*Trametes versicolor and Trichoderma harzianum, Glomus deserticola* and *G. claroideum*	*Eucalyptus globulus* Labill.	Arsenic (As)	Increased the shoot and root dry weight, and chlorophyll content	[52]
*Trichoderma* sp. *PDR1-7*	*Pinus sylvestris* L.	lead	Removed heavy metals from mine-tailing soil extract media	[65]
*Penicillium aculeatum PDR-4 and Trichoderma* sp. *PDR-16*	*Sorghum*, *Sudangrass*	As, Cu, Pb and Zn	Caused growth and As, Pb, and Zn uptake	[66]
Arbuscular mycorrhizal fungi	*Bread wheat*	Cd	Improved nitrogen and phosphorus nutrition, and the immobilization of Cd	[67]
*Cadophora*, *Leptodontidium*, *Phialophora* and *Phialocephala*	N/A	Cd, Pb and Zn	Promoted plant growth, metabolite production, and metal tolerance	[68]
*Aspergillus fumigatus, Rhizopus* sp., *Penicillium radicum* and *Fusarium proliferatum*	*Lactuca sativa* L.	Cr-VI to Cr-III	Detoxified up to 95% of Cr extracellularly	[69]
*Trichoderma atroviride F6*	*Brassica juncea* L. *Coss.* var. *foliosa Bailey*	Cd, Ni	Caused an 110%, 40%, and 170% increase in fresh weight	[53]
*Glomus geosporum*	*Aster tripolium* L.	Cd and Cu	Enhanced Cd and Cu root uptake and accumulation	[70]
Arbuscular mycorrhizal fungi	*Erato polymnioides*	Hg	Increased Hg accumulation	[71]
*Claroideoglomus etunicatum*	*Zea mays* L.	La and Cd	Caused metal uptake and transport	[72]
Arbuscular mycorrhizal fungi (AMF)	*Solanum melongena* L.	Pb, Cd, and As as Pb (NO_3_)_2_, CdCl_2_·5H_2_O, and As_3_S_2_	Improved growth, biomass, and the antioxidative defense response	[73]
Arbuscular mycorrhiza (AM)	*Trifolium pratense* L.	Zn	Increased Zn uptake and root accumulation, enhanced plant growth and P nutrition, and alleviated Zn toxicity	[74]
Arbuscular mycorrhizal fungi	*Populus cathayana*	Pb	Increased P uptake, antioxidant enzyme activity, and Pb accumulation	[75]
*Glomus intraradices* (AH01)	*Oryza sativa* L.	Arsenite	Decreased arsenite uptake and immobilized arsenite in rice roots, preventing translocation to shoots	[76]
*Glomus intraradices (AH01)*	*Oryza sativa* L.	Arsenate	Increased OsPT11 expression, enhanced P concentration and biomass, decreased arsenate concentration and uptake in rice, and raised the P/As molar ratio	[77]
AMF Funneliformis mosseae (Fm) or Rhizophagus intraradices (Ri)	*Oryza sativa* L.	Cd	Reduced rice Cd uptake by altering Cd transporter expression	[78]
*Fusarium* sp. *CBRF44, Penicillium* sp. *CBRF65, and Alternaria* sp. *CBSF68*	*Brassica napus*	Pb and Cd	Increased biomass and metal extraction	[79]
*Aspergillus fumigatus, Aspergillus niger, Fusarium equiseti, Fusarium chlamydosporum, Paecilomyces lilacinus, Trametes versicolor, Penicillium cataractum, Perenniporia subtephropora, Daldinia starbaeckii, Antrodia serialis, Cerrena aurantiopora, Phanerochaete concrescens* and *Polyporales species.*	*Prosopis juliflora* Sw.	As,Cr,Cu,Fe,Mn,Ni, Pb,Zn	Enhanced biomass, increased root and shoot lengths, elevated carotenoids and chlorophyll, increased levels of L-phenylalanine and L-leucine, increased heavy metal accumulation, upregulated antioxidant genes, improved growth and metal tolerance, and served as an effective bioremediation strategy	[44]
*Trichoderma harzianum*	*Amaranthus hypochondriacus* L.	Cd and Zn	Promoted phytoremediation of Cd and Zn and enhanced the prevalence of heavy metal resistant genes (MRGs) and antibiotic resistance genes (ARGs), MRGs were influenced by available Zn and Cd, and MRGs were linked to specific bacterial hosts	[80]

## 4. Plant–Microbe Synergism in Soil Recovery

The alliance of microorganisms and phytoremediation has acknowledged increasing consideration and is a well thought out, promising remediation technology for improving the adsorption and modification of heavy metals [81]. Plant growth-promoting bacteria (PGPB)-assisted phytoremediation has been used to enhance the phytoremediation of HM-contaminated soils [82]. Plant growth-promoting bacteria (PGPB) enhance plant growth through various mechanisms, both directly and indirectly. This mode of action includes nitrogen fixation, where bacteria convert atmospheric nitrogen into a plant-usable form, and phosphate solubilization, which makes phosphorus more accessible for plant uptake. Furthermore, PGPB emit siderophores that chelate and transport iron to the plant, phytohormones such as auxins and gibberellins that control growth and development, and ACC deaminase, an enzyme that alleviates plant stress by breaking down the ethylene precursor ACC, thus promoting healthy plant growth [83]. The indirect elevation of plant development by PGPB usually defends plants against various pathogens or recovers their resistance to environmental stresses such as drought, salt, heavy metals, and organic contaminants [84]. Among these growth-promoting microbes, some strains have been found to be tolerant to high concentrations of HMs [85]. Apart from modifying plant cell metabolism to drive growth, these bacteria can provide additional benefits to host plants, such as mitigating the harmful effects of heavy metals (HMs) and permitting plants to withstand high HM concentrations. With respect to this, heavy metal-tolerant plant growth-promoting bacteria (HMT-PGPB) have a substantial probability for phytoremediation by increasing plant survival and growth in HM-contaminated soils [86]. Rhizospheric and internal plant microbes can stimulate biofilm production around the root zones of host plants. Biofilm buildup, which comprises the accumulation of microbial cells in a self-produced extracellular matrix, augments the resilience of these microbes in intimate environments containing toxic pollutants. This defensive biofilm layer not only helps microbes adhere to root surfaces but also forms a microenvironment that protects them from stressors, such as heavy metals and other contaminants. Additionally, biofilms can enable nutrient exchange between microbes and plants, enhance pollutant degradation, and advance the overall health and growth of host plants under adverse conditions. *Kocuria flava* and *Bacillus vietnamensis*, when applied as an inoculum, as well as Kocuria flava and Bacillus vietnamensis, not only promoted the growth of rice seedlings but also decreased As uptake and accumulation in plants [87]. *Pseudomonas* sp. H13 and *Brevundomonas* sp. H16 initiate the production of extracellular polysaccharides and inorganic labile sulfides and strengthen biofilm formation, thereby significantly improving the removal efficiency of Cu^2+^, Zn^2+^, Cd^2+^, and Pb^2+^ [88]. With the integration of metal-tolerant bacteria isolated from polluted soil with plant growth promotion (PGP) potential, *Brassica juncea* and *Lupinus albus* phytoaccumulation increased up to 85% for As and up to 45% for Hg [89]. Biofilm communities are thus proficient in the sorption and metabolism of organic pollutants and heavy metals through a well-controlled expression pattern of genes governed by quorum sensing [90]. The main policy for improving the phytoremediation of contaminated soils is to enhance the expression of specific genes using CRISPR technology, thereby boosting the production of metal-binding proteins such as metallothioneins (MT) and phytochelatins (PC), metal transporter proteins (from the MATE, CDF, HMA, ZIP, and YSL families) [91], plant growth hormones (such as auxins, cytokinins, and gibbrellic acids) [92], and root exudates [93]. CRISPR is a powerful gene-editing tool that allows for precise modifications of an organism’s DNA, making its application in areas such as soil remediation and cleanup highly valued. Some examples include the targeted activation or repression of genes responsible for metal uptake, transport, and sequestration.

Numerous studies since the early 2000s have discovered that specific plant and bacterial genes, when combined with the target plant genomes, might enhance phytoremediation abilities [94]. For example, expression of the NAS1 gene (responsible for encoding the enzyme nicotianamine synthase-1) in tobacco and *Arabidopsis* plants resulted in enhanced metal tolerance for Cd and Zn [95]. Likewise, some genes, including *czcD*, are important for bacteria to purify Cd^2+^ [96]. Some of the metal resistance genes are listed in Table 3.

**Table 3 microorganisms-12-01945-t003:** Exploration of metal-resistant genes from bacteria.

Species Name	Resistant Genes	HM	References
*Citrobacter*, *Desulfocurvus* and *Stappia*	*czcA, czcB, czcC*	Cadmium	[97]
*Caenispirillum, Halomonas, Stappia, Thauera*	*ctpA, copZ, copR* and *copB*	Cadmium	[98]
*Serratia marcescens* CCMA 1010	*zntR gene*	Pb^2+^	[99]
*Pseudomonas, Escherichia coli* and *Staphylococcus aureus*	*merR, merD*	Hg^2+^	[100]
*Sporosarcina ginsengisoli*	*copK*	As(III)	[101]
*Georgenia* sp. SUBG003	*czcD*	Cobalt/zinc/cadmium resistance protein	[102]
*Cupriavidus metallidurans*	*pbrR* and *its derivatives, pbrR2* and *pbrD*	Lead	[103]
*Escherichia coli*	*Metallothionein (MT)*	Cd	[104]
*Thiobacillus, Hydrogenophaga* and *Flavihumibacter*	*aioA, arsC, arrA* and *soxB genes*	Arsenic	[105]
*Microbacterium paraoxydans*	*arsR, arsB, arsC, acr1, acr2* and *acr3*	Arsenic	[106]
*Pseudomonas putida* ARS1	*aio, arr,* and *arsM*	Arsenic	[107]
*Desulfurella* and *Clostridium*	*asrA* and *arsB*	Arsenic	[108]
*Acinetobacter* sp. (ADHR1)	*chrR*	Cr (VI)	[109]
*Alphaproteobacteria Xanthobacter autotrophicus Py2*	*mer1* and *mer2*	Mercury (Hg)	[110]
*Bacillus megaterium*	*merA* and *merB*	Mercury (Hg)	[111]
*Pseudomonas putida*	*mer73*	Mercury (Hg)	[112]
*E. coli K12*	*nikA, nikE, nikC, rcnA* and *nikB,*	Ni^2+^	[113]
*Alcaligenes xylosoxydans, Ralstonia metallidurans* and *Helicobacter mustelae*	*nccA, cnrA* and *cznA*	Ni^2+^	[113]
*Arthrobacter rhombi* AY509239, *Clavibacter xyli* AY509235, *Microbacterium arabinogalactanolyticum* AY509226, *Rhizobium mongolense* AY509209 and *Variovorax paradoxus* AY512828	*czc, chr, mer* and *ncc*	Arsenate, cadmium, chromium, zinc, mercury, lead, cobalt, copper, and nickel	[114]
*Pseudomonas aeruginosa* JP-11	*cad* operon*,* and *czc* operon	Cadmium	[115]
*Staphylococcus aureus*	*cadB*	Cadmium	[116]
*Staphylococcus lugdunensis*	*cadX*	Cadmium	[117]
*Ochrobactrum tritici*	*chrBACF*	Chromium	[118]
*Arthrobacter* sp.	*chrJ, chrK, and chrL*	Chromium	[119]
*Escherichia coli*	*cueO*	Copper	[120]
*Pseudomonas fluorescens*	*copRSCD operon*	Copper	[121]
*Bacillus subtilis*	*ycnJ*	Copper	[122]
*Helicobacter pylori*	*hpcopA and hpcopP*	Copper	[123]
*C. metallidurans*	*pbrU*	Lead	[124]
*Pseudomonas aeruginosa strain* WI-1	*bmtA*	Lead	[125]
*Acidithiobacillus ferrooxidans*	*merC*	Mercury	[126]
*Cupriavidus (Ralstonia) metallidurans*	*cnrCBA*	Nickle	[127]
*Achromobacter xylosoxidans* 31A	*nre*	Nickle	[128]
*Escherichia coli*	*yohM*	Nickle	[129]
*Escherichia coli*	*NiCoT* efflux gene (*rcnA*)	Nickle and Cobalt	[130]
*Escherichia coli*	*ppk*	Mercury	[131]

Heavy metal transport can be influenced by the interaction between host plants and diverse AM fungal isolates, and the contact of heavy metals with other metals [132]. Arbuscular mycorrhizal fungi (AMF) are obligate symbionts of an extensive array of plants. Their contribution as adjuncts in phytoremediation must be alongside their plant hosts, unlike fungi such as *Trichoderma*, which can independently colonize the soil and be cultivated and/or applied in soil remediation interventions without an associated host.

AM fungi may enhance stress tolerance in polluted soils by trapping heavy metals in their extraradical hyphae and plant root systems [133]. Arbuscular mycorrhizal (AM) fungi are crucial for enhancing the accumulation of glomalin-related soil proteins, organic matter, and organic carbon. This is achieved through several mechanisms. AM fungi contribute to stabilizing and forming soil aggregates by producing glomalin, a glycoprotein that binds soil particles together, improving soil structure, and enhancing organic matter content. This stabilization process can alter the particle size distribution, leading to more structured and less erodible soil. Additionally, by forming symbiotic relationships with plant roots, AM fungi enhance plant growth and nutrient uptake, increasing the input of organic matter into the soil as root exudates and decaying plant material. AM fungi can also mitigate metal toxicity in areas contaminated with heavy metals by sequestering metals in their hyphal networks and reducing their bioavailability, thereby further influencing soil health and structure [134]. A cyclin, specifically SiPHO80, within the protein family, may be crucial for maintaining inorganic phosphate balance and managing tolerance to heavy metal stress in *Serendipita indica*, an osmotolerant AM fungal species [135]. Therefore, *S. indica* has been suggested as a potential biofertilizer. We can also benefit from their role in heavy-metal phytoextraction. AMF-assisted remediation of metal contamination involves several modes of action, such as increasing photosynthetic capability, facilitating nutrient absorption, accelerating biochemical and enzymatic activities, restricting heavy metal uptake, altering soil pH, and immobilizing or compartmentalizing heavy metals in plant organs and fungal structures, thus enhancing plant performance [136]. AMF (e.g., *Glomus, Rhizophagus*) boost plant growth and resilience to heavy metal stress for effective phytoremediation [137].

## 5. Engineering the Meta-Organism

The meta-organism is the plant and its associated microbiome; therefore, it has been proposed that engineering could be a viable strategy to make phytoremediation processes more efficient. This proposal was made by Thijs et al. [138] a few years ago, where, based on different studies, they described four main strategies for engineering a meta-organism. Phytoremediation, traditionally focused on selecting plants with fast growth, high biomass, and hyperaccumulation capabilities, often overlooks the influence of plants on their associated microbiomes, which is crucial for the success of phytoremediation [1,6]. The selection of plant species significantly affects the structure and function of microbial communities, making it essential to modify plant-associated microbial communities. Therefore, integrating microbiome functions and promoting specific microbial assemblies can improve both phytoremediation and biomass yields. Practical strategies include evaluating the effects of plants on microbiomes and implementing interventions to enhance the microbiomes and phytoremediation outcomes. This approach is termed “Microbiome-based plant selection” [2].

Another strategy proposed by the authors was the manipulation of root exudates. Known previously as “top-down” engineering by Orozco-Mosqueda and colleagues, this strategy involves manipulating the plant genome to produce exudates that increase or stimulate the abundance of beneficial microbial groups [139]. Plant molecules, such as strigolactones, flavonoids, and cutin monomers, act as signals detected by rhizosphere microorganisms. Similarly, the rhizobiome influences the plant transcriptome and metabolism, thereby impacting its overall fitness. This interplay involves mutual effects, where both plants and their rhizobiomes interact through the secretion and detection of signals in the rhizosphere. Despite significant efforts to understand the chemicals coordinating symbiotic interactions in the rhizosphere, redirecting rhizosphere microbial communities remains challenging owing to the dynamic nature and variability of rhizodeposits, which are influenced by plant species, physiological stage, neighboring plants, soil characteristics, soil contaminants, and microbial community context [11]. Future research should further explore this topic, as it holds promise for manipulating rhizomicrobiomes through plant root exudates.

The third strategy involves modification or perturbation of the driving forces. Competition can favor the selection of beneficial plant growth promoters (PGPs) and degrading microorganisms in the rhizosphere. Metagenomic data have shown that many microbial genes required for phytoremediation are already present in the environment. However, this is sometimes insufficient for achieving high biodegradation activity. Contaminant degradation results from various competitive interactions, including interference competition and resource exploitation, as well as cooperative interactions, such as coexistence, mutualism, and symbiosis, which affect partners spatially and temporally. Identifying and understanding these interactions between the host plant and its microbiome are crucial for optimizing the organism [138]. Improving our understanding of these relationships presents challenges, including designing studies to determine whether interactions are direct or indirect, and addressing issues related to statistical assessment and testing of the dynamics, feedback, and uncertainties in host–microbiome relationships.

Finally, engineering rhizomicrobial catabolic capabilities have been proposed [138]. This approach addresses cases in which the abundance of degrading traits in the surrounding soil is low. Increasing the frequency of beneficial plant growth-promoting and degrading microorganisms can be achieved through enrichment or inoculation, thereby enhancing net immigration from the environment through competitive interactions [2]. Additionally, preselecting highly colonizing strains with competitive capabilities in multiple environments would increase the confidence in achieving a bioinoculant with better field results. For example, under saline conditions in agricultural soils, certain mycobacterial agents in a bioinoculant should have strategies to survive such stress, such as biofilm production, membrane component modification, and osmoprotectant production.

## 6. Assessment of Perspectives

This study has emphasized the dynamic role of soil microorganisms, predominantly bacteria and fungi, in supplementing the proficiency of phytoremediation for agronomic soils polluted with heavy metals. Although plants have intrinsic capabilities to absorb, degrade, and stabilize pollutants during phytoextraction, phytodegradation, and phytostabilization, their microbial cohorts play a secondary role in enhancing these methods. These microbes, dwelling in the rhizosphere and endosphere, increase plant tolerance to heavy metals, activate and immobilize contaminants, and enable nutrient uptake, thereby promoting plant growth and overall phytoremediation efficiency [140].

Numerous research directions have emerged in this regard. A crucial priority is to attain a comprehensive understanding of the diverse microbial ecosystems connected with specific plant species and their roles in heavy metal remediation. Accomplishing this demand requires advanced molecular techniques for describing these microbial populations, such as metagenomics, metatranscriptomics, and metaproteomics. These omics sciences are fundamental to unveiling novel mechanisms of plant–microbe interactions and genes upregulated in the presence of heavy metals, particularly in agricultural soils, etc. [141]. Additionally, recognizing and isolating new microbial strains with improved metal resistance and plant growth-promoting capabilities, especially those specializing in heavy metals and plant species, is vital. Reconnoitering the potential of extremophiles adapted to environments with high heavy metal contamination is also essential to continue investigating and taking advantage of these groups of prokaryotes, particularly those that exhibit beneficial interactions with plant crops [142]. Furthermore, genetic engineering/genome modification approaches can be leveraged to boost the phytoremediation effectiveness of plants and their symbiotic microorganisms. This involves incorporating genes that enhance metal tolerance, increasing the production of chelating agents, or modifying microbial metabolic pathways to improve metal transformation efficiency. These are promising options, with their interactions and molecular mechanisms only beginning to be understood for enhancing phytoremediation in agricultural soils, particularly through the use of soil-adapted and endemic microorganisms.

## 7. Conclusions

Further research is important to improve the application of microbial consortia in phytoremediation, considering variables such as soil type, pollutant concentrations, and plant species. Inspecting the interactions between microbial strains and their community impact on pollutant degradation can reveal synergistic effects that augment remediation efficiency. Understanding the genetic and metabolic responses of microbes and plants to contaminants can lead to the development of more targeted and resilient phytoremediation strategies. By exploring these factors, we can optimize the use of microbial consortia to address persistent heavy-metal contamination and improve soil health and fertility [143,144]. In turn, this will support advancements in food security, promote sustainable agricultural practices, and contribute to overall environmental restoration.

## Figures and Tables

**Figure 1 microorganisms-12-01945-f001:**
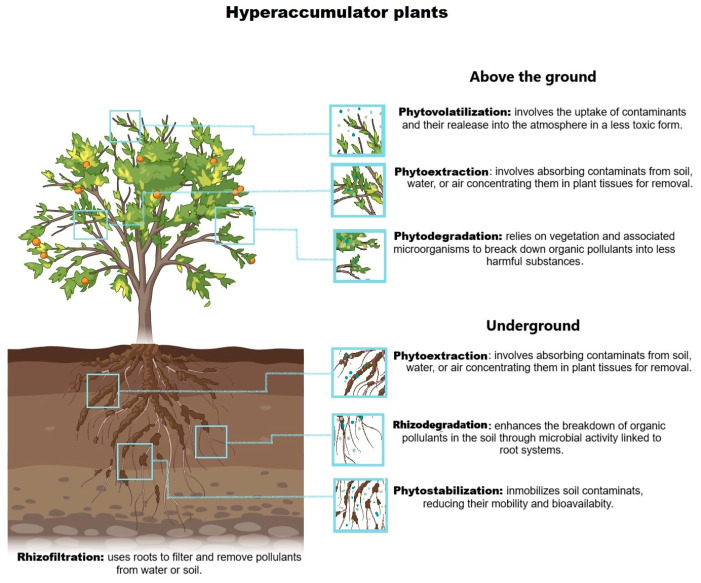
The mechanism of action for microbial-assisted phytoremediation in soils contaminated with heavy metals. This figure depicts various phytoremediation technologies used to address environmental contamination [11,20,21].

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
