# Peer review of "Microbial Contributions to Heavy Metal Phytoremediation in Agricultural Soils: A Review"

_microorganisms, 2024, doi:10.3390/microorganisms12101945_

Round 1

Reviewer 1 Report

Comments and Suggestions for Authors

Dear authors,

 The review provides a well-written and extensive assessment of microorganisms' function in improving the phytoremediation of heavy metals in agricultural soils. It successfully emphasizes the mutually beneficial relationships between plants and microorganisms and offers useful insights into future research directions. The work makes an important contribution to the field of environmental biotechnology, especially in terms of sustainable agriculture and soil remediation. Future assessments could improve by integrating more practical case studies and addressing this field's economic and ethical considerations. So, the review is accepted in the present form.

Thank you very much.

Author Response

Thank you very much for your comments! A new revised copy is attached.

Reviewer 2 Report

Comments and Suggestions for Authors

Dear Editor

Many thanks for considering me as a potential reviewer for the article "Microbial Contributions to Heavy Metal Phytoremediation in Agricultural Soils: An Updated Review" The article is undoubtedly well-structured, well-presented and well-written. However, I have several observations that should be considered before proceeding further.

My observations are as follows.

Major considerations

·       English is not fine, please do extensive English editing by a native English speaker,

·       Tables are confusing, in terms of metal names i.e. metal names are not uniform, please use the full name and/or abbreviations in all tables.  Another, please take care of smaller and capital letters too,

·       Some scientific names lacking authority names, more specifically in tables, please consider,

·       Lines 171-176 this portion is too confusing (bold text), also please cite this information, accordingly ‘The explanation of plant-advantageous bacteria, largely metal resistant strains, and the contemplation of their mechanisms to augment plant growth and metal tolerance (discussed in this book chapter) is pivotal to undertaking phytoremediation-enhancing tactics. Table 1 presents the number of studies directed in this context using PGPR and PGPE. Plant health in metal-polluted soils is upgraded in two ways by these bacteria: First, through the direct production of plant growth beneficial products together with solubilization/transformation of mineral nutrients (e.g.: phosphate, nitrogen, and potassium),’ The source [29] you cited is a review article not a book.

·       Figure 1, (1) Would be more interesting if you present the figure in a way of up (a) and below (b) ground and then explain the various approaches you mentioned. (2) what is the source of the figure, please mention and/or cite it)

·        

Minor comments

·       Line-11, please re-phrase this line ‘However, during the cleanup process, plants may not act alone and could receive help from their associated microorganisms’

·       Line-364 ‘Cd, Zn’ should be ‘Cd and Zn’ please do the said throughout MS,

·       Line-401, please cite this paragraph text ‘Engineering the metaorganism’,

·       Line-16, Would be more interesting if you mention those plants and their microbes ‘Remarkably, some plant species are known for their hyper-accumulative traits in synergy with their microbial partners and can successfully mitigate heavy metal pollutants.’

·       Line-60 gives few examples ‘Hyperaccumulator plant species’,

·       Lines 66-67, please mention some potential species ‘Some bacteria and fungi are also extremely tolerant to manifold heavy metals so that they can endure in rhizospheric environments’.

Comments on the Quality of English Language

Dear Editor/Authors,

Several issues regarding the English language have been detected, I will suggest the authors do extensive English editing either by an English native professional in the field and/or by a senior Prof. 

Thanks!

Author Response

RESPONSE TO REVIEWERS

REVIWER 1

Dear authors,

 The review provides a well-written and extensive assessment of microorganisms' function in improving the phytoremediation of heavy metals in agricultural soils. It successfully emphasizes the mutually beneficial relationships between plants and microorganisms and offers useful insights into future research directions. The work makes an important contribution to the field of environmental biotechnology, especially in terms of sustainable agriculture and soil remediation. Future assessments could improve by integrating more practical case studies and addressing this field's economic and ethical considerations. So, the review is accepted in the present form.

Thank you very much.

RESPONSE: Thank you very much for your kind words and review of our work.

REVIEWER 2

Many thanks for considering me as a potential reviewer for the article "Microbial Contributions to Heavy Metal Phytoremediation in Agricultural Soils: An Updated Review" The article is undoubtedly well-structured, well-presented and well-written. However, I have several observations that should be considered before proceeding further.

My observations are as follows.

Major considerations

  • English is not fine, please do extensive English editing by a native English speaker,

RESPONSE: Thank you for drawing our attention to this issue. The manuscript has been edited by the MDPI English editing services.

  • Tables are confusing, in terms of metal names i.e. metal names are not uniform, please use the full name and/or abbreviations in all tables.  Another, please take care of smaller and capital letters too,

RESPONSE: You are right. We corrected all names with errors. We also corrected some strain names and species abbreviations. Thank you!

  • Some scientific names lacking authority names, more specifically in tables, please consider,

RESPONSE: Could you be so kind as to indicate which ones we should include? This could be very relative, but we tried to include good reputed journals instead of looking at the names, because we do not know them.

  • Lines 171-176 this portion is too confusing (bold text), also please cite this information, accordingly ‘The explanation of plant-advantageous bacteria, largely metal resistant strains, and the contemplation of their mechanisms to augment plant growth and metal tolerance (discussed in this book chapter) is pivotal to undertaking phytoremediation-enhancing tactics. Table 1 presents the number of studies directed in this context using PGPR and PGPE. Plant health in metal-polluted soils is upgraded in two ways by these bacteria: First, through the direct production of plant growth beneficial products together with solubilization/transformation of mineral nutrients (e.g.: phosphate, nitrogen, and potassium),’ The source [29] you cited is a review article not a book.

RESPONSE: Our apologies for this mistake, which was corrected. We deleted the phrase: (discussed in this book chapter). Thanks.

  • Figure 1, (1) Would be more interesting if you present the figure in a way of up (a) and below (b) ground and then explain the various approaches you mentioned. (2) what is the source of the figure, please mention and/or cite it)

RESPONSE: Great suggestion! Done. We hope you like it the new Figure, thanks.

  •  

Minor comments

  • Line-11, please re-phrase this line ‘However, during the cleanup process, plants may not act alone and could receive help from their associated microorganisms’

RESPONSE: Modified as suggested.

  • Line-364 ‘Cd, Zn’should be ‘Cd and Zn’ please do the said throughout MS,

RESPONSE: Modified as suggested.

  • Line-401, please cite this paragraph text ‘Engineering the metaorganism’,

RESPONSE: Modified as suggested. We included citations 1, 2, 6 , 11 and 138 in the subheading.

  • Line-16, Would be more interesting if you mention those plants and their microbes ‘Remarkably, some plant species are known for their hyper-accumulative traits in synergy with their microbial partners and can successfully mitigate heavy metal pollutants.’

RESPONSE: Thank you for your suggestion, however the length of the abstract does not allow it, but the names are mentioned in the work.

  • Line-60 gives few examples ‘Hyperaccumulator plant species’,

RESPONSE: Modified as suggested, thank you for your observation! We included

Nymphaea nymphaea, Bornmuellera tymphaea, Pteris vittate, Sedum alfredii and Thlaspi caerulescens, as examples.

  • Lines 66-67, please mention some potential species ‘Some bacteria and fungi are also extremely tolerant to manifold heavy metals so that they can endure in rhizospheric environments’.

RESPONSE: Modified as suggested, thank you!

Dear Editor/Authors,

Several issues regarding the English language have been detected, I will suggest the authors do extensive English editing either by an English native professional in the field and/or by a senior Prof. 

Thanks!

RESPONSE: Dear reviewer, thank you for your kind comments, particularly we would like to let you know that our revised work was edited by the professional editing services of the company EDITAEGE. We have submitted only that version with the English changes highlighted. This version includes the response to your comments and those from other reviewers. However, we could not submit both versions. Thank you again. Best wishes, Gustavo Santoyo 

Reviewer 3 Report

Comments and Suggestions for Authors

The manuscript microorganisms-3211404 describes the importance of microorganisms (bacteria and fungi) for the success of heavy metal phytoremediation. The authors use "An update review" in the title, however, the manuscript does not provide information on the period of time for which the authors wanted to make an update review. My main comment on the text of the manuscript is that when describing the material, the authors often give preference to old articles rather than to publications from recent years (2023-24), of which there are also many, including those describing the genomes and genes involved in resistance to heavy metals. There are also many errors in the manuscript (see below). For these reasons, I think the manuscript needs a major revision.

Remarks:

1. Line 24: In the manuscript text (line 402) and in reference 138, the word "Metaorganism" is written as one word.

2. Line 38: "5 g/cm3"? Heavy metals? That's a lot! The reference Chen et al. does not provide this information.

3. Line 75: "such as Arbuscular Mycorrhizal Fungi (AMF)" would be more correct "including Arbuscular Mycorrhizal Fungi (AMF)".

4. Lines 113 and 116: The plant names are misspelled. The correct names are: "Bornmuellera tymphaea (Hausskn.) Hausskn.", "Noccaea tymphaea (Hausskn.) F.K.Mey.", and "N. tymphaea". And from here on in the manuscript, incomplete plant names (with authors) are given: for example, lines 125, 191-192, 196, Tables 1 and 2, and so on.

5. Line 114: What are "ABi and NBi"?

6. Line 119: For acetate, the unambiguous spelling is "CH3COO".

7. Lines 138-140: What is the role of PGPB in phytovolatization?

8. Line 146: "total hemofiltration efficacy" - Why "hemo-"?

9. Line 163-165: For the given list of bacterial genera, reference [19] is insufficient.

10. Lines 171-174: This sentence is an unfortunate modification of a sentence from Alves et al., 2022, which is not in the References section. What is "(discussed in this book chapter)"?

11. Line 174: What is "PGPE"?

12. Line 185: "Serratia sp. K120" would be correct.

13. Lines 205-211: Why is this passage relevant to phytoremediation? Azospirillum spp. are not heavy metal-resistant bacteria. Additionally, strain Sp245 has been the type strain for Azospirillum baldaniorum since 2020. This passage should be removed or modified and clarified.

14. Line 219: What is "PGPBE"?

15. Table 1: First, the heading of column 2 is "Strain name", but only 8 strains have strain names, while the remaining bacteria only have species names. Please provide strain names for all bacteria. Second, the table provides data from publications for the period 2015-2022. Moreover, only 2 references are from 2021, and only 1 reference is from 2022. The remaining references are from 2018 and earlier. This should be commented on by the authors in the text of the manuscript. Why did the authors limit themselves to reviewing publications up to 2022?

16. Tables 2 and 3 lack uniformity in the writing of heavy metals and genes, respectively.

17. Line 367: Table 3 only lists bacterial genes, so “metal-resistant genes from bacteria” would be correct.

18. What is the principle of data placement in Table 3? I did not see any order for bacteria, genes, or heavy metals.

19. Lines 444-453: Provide references for this information.

20. Line 481: Provide references for “synthetic communities”.

Author Response

Dear Reviewer 3,

Thank you for taking the time to review our manuscript. Your suggestions and improvements were very welcome. We have another version of the manuscript, edited by a native English speaker from Editage, which we could not upload here. However, this version has been submitted to the Editors of Microorganisms. Attached you can see a version with highlighted changes, some of them to answer your comments.

Best wishes,
Gustavo Santoyo

We are answering as follows:

The manuscript microorganisms-3211404 describes the importance of microorganisms (bacteria and fungi) for the success of heavy metal phytoremediation. The authors use "An update review" in the title, however, the manuscript does not provide information on the period of time for which the authors wanted to make an update review. My main comment on the text of the manuscript is that when describing the material, the authors often give preference to old articles rather than to publications from recent years (2023-24), of which there are also many, including those describing the genomes and genes involved in resistance to heavy metals. There are also many errors in the manuscript (see below). For these reasons, I think the manuscript needs a major revision.

RESPONSE: Thank you for reviewing our work and all your suggestions. The title was modified as suggested.

Remarks:

  1. Line 24: In the manuscript text (line 402) and in reference 138, the word "Metaorganism" is written as one word.

RESPONSE: You are right; we corrected the term 'Meta-organism' throughout the manuscript. This term is widely used in the literature. Thank you.

  1. Line 38: "5 g/cm3"? Heavy metals? That's a lot! The reference Chen et al. does not provide this information.

RESPONSE: Thank you very much for your observation, the sentence was corrected as follows: Current statistical analysis revealed that a substantial proportion of agronomic lands worldwide have been affected by heavy metal pollution, with cadmium (Cd), copper (Cu), chromium (Cr), mercury (Hg), lead (Pb), zinc (Zn), and nickel (Ni) mainly tainting arable soils [4]. According to Chen et al., the metalloid arsenic (As) is frequently included in this category due to its analogous chemical properties and environmental behavior [5].

  1. Line 75: "such as Arbuscular Mycorrhizal Fungi (AMF)" would be more correct "including Arbuscular Mycorrhizal Fungi (AMF)".

RESPONSE: Modified as suggested, thank you!

  1. Lines 113 and 116: The plant names are misspelled. The correct names are: "Bornmuellera tymphaea(Hausskn.) Hausskn.", "Noccaea tymphaea(Hausskn.) F.K.Mey.", and "N. tymphaea". And from here on in the manuscript, incomplete plant names (with authors) are given: for example, lines 125, 191-192, 196, Tables 1 and 2, and so on.

RESPONSE: Thank you very much for your suggestion and we know you are right but we prefer to include only the species names, this is also a correct and widely used form in the literature (e.g. Ly, S.N., Echevarria, G., Aarts, M.G.M. et al. Physiological responses of the nickel hyperaccumulator Bornmuellera emarginata under varying nickel dose levels and pH in hydroponics. Plant Soil (2024). https://doi.org/10.1007/s11104-024-06777-6).

  1. Line 114: What are "ABi and NBi"?

RESPONSE: THANK YOU FOR YOUR OBSERVATION. They are a label of the experiment. We corrected and modified the text to give clarity: ¨ The association of Bornmuellera nymphaea - Noccaea nymphaea (NB) and Bornmuellera tymphaea - Alyssum murale (AB) inoculated with PGPR…¨

  1. Line 119: For acetate, the unambiguous spelling is "CH3COO".

RESPONSE: Modified as suggested, thank you!

  1. Lines 138-140: What is the role of PGPB in phytovolatization?

RESPONSE: The short answer is NONE but thank you for your question. We modified and corrected the information as follows: ¨ Phytovolatization has made a notable contribution to the detoxification of heavy metals. It remained a key mechanism in arsenic (As) decontamination during the bioremediation process involving Arundo donax L. and the amicrobial assistance of epigenetic changes by a PGPR consortium. The PGPR was constituted of two strains of Stenotrophomonas maltophilia and one Agrobacterium sp. strain. The authors analyzed the methylation sensitive amplified polymorphisms (MSAP) of A. donax leaves and detected potential changes by the interaction with As and the bacterial consortium. The presence of PGPB did not modify the percentage of arsenic volatilized or induced significant plant growth biomass, but the work demonstrated that phytovolatilization is a robust mechanism for reducing arsenic concentrations in contaminated environments [25].¨

  1. Line 146: "total hemofiltration efficacy" - Why "hemo-"?

RESPONSE: A mistake corrected. It is filtration only.

  1. Line 163-165: For the given list of bacterial genera, reference [19] is insufficient.

RESPONSE: Modified as suggested, thank you! Included [1, 2, 10, 12, 13, 19].

  1. Lines 171-174: This sentence is an unfortunate modification of a sentence from Alves et al., 2022, which is not in the References section. What is "(discussed in this book chapter)"?

RESPONSE: Sorry about it. The sentence was deleted.

  1. Line 174: What is "PGPE"?

RESPONSE: (Plant growth-promoting endophytes) the definition was added.

  1. Line 185: "Serratiasp. K120" would be correct.

RESPONSE: Modified as suggested.

  1. Lines 205-211: Why is this passage relevant to phytoremediation? Azospirillumspp. are not heavy metal-resistant bacteria. Additionally, strain Sp245 has been the type strain for Azospirillum baldaniorumsince 2020. This passage should be removed or modified and clarified.

RESPONSE: THANK YOU. The species name was corrected and we added the following paragraph to give clarity: ¨ This result shows that although there are naturally non-resistant strains to heavy metals like A. baldaniorum, there are others that can tolerate broader ranges and could be candidates for inoculation in plants, thereby stimulating their fitness and growth, and in turn, indirectly improving remediation processes¨.

  1. Line 219: What is "PGPBE"?

RESPONSE: The sentence was modified as follows: ¨ Table 1. Relevant studies demonstrating the role of plant growth-promoting rhizobacteria and endophytic bacteria in the phytoremediation of metal-contaminated soils, including those used for agriculture.¨

  1. Table 1: First, the heading of column 2 is "Strain name", but only 8 strains have strain names, while the remaining bacteria only have species names. Please provide strain names for all bacteria. Second, the table provides data from publications for the period 2015-2022. Moreover, only 2 references are from 2021, and only 1 reference is from 2022. The remaining references are from 2018 and earlier. This should be commented on by the authors in the text of the manuscript. Why did the authors limit themselves to reviewing publications up to 2022?

RESPONSE: Thak you for your comment. We corrected legend title of the table, the scientific and strain names, adding sp. to those unidentified strains. The title was also modified ¨Species/Strain¨. About the works, we consider those are relevant and not reviewed in other works, to the best of our knowledge.

  1. Tables 2 and 3 lack uniformity in the writing of heavy metals and genes, respectively.

RESPONSE: You are right. This was also indicated by another reviewer. The Table was corrected as suggested.

  1. Line 367: Table 3 only lists bacterial genes, so “metal-resistant genes from bacteria” would be correct.

RESPONSE: MODIFIED AS SUGGESTED.

  1. What is the principle of data placement in Table 3? I did not see any order for bacteria, genes, or heavy metals.

RESPONSE: None.

  1. Lines 444-453: Provide references for this information.

RESPONSE: Modified as suggested.

  1. Line 481: Provide references for “synthetic communities”.

RESPONSE: Reference added: Santoyo, G.; Guzmán-Guzmán, P.; Parra-Cota, F.I.; Santos-Villalobos, S.d.l.; Orozco-Mosqueda, M.d.C.; Glick, B.R. Plant Growth Stimulation by Microbial Consortia. Agronomy 202111, 219. https://doi.org/10.3390/agronomy11020219

Round 2

Reviewer 2 Report

Comments and Suggestions for Authors

Dear Authors/Editor

Many thanks for considering my queries and kind responses. The article is undoubtedly much better now, however, still a few important corrections I think are mandatory to address.

·       What is the source of the figure, please mention,

·        Previous comment (Some scientific names lacking authority names, more specifically in tables, please consider,

RESPONSE: Could you be so kind as to indicate which ones we should include? This could be very relative, but we tried to include good reputed journals instead of looking at the names, because we do not know them.)

  • As per your/authors' request, I am trying to explain this issue, please check table No. 1, how the plant names are mentioned (DOI: 10.3390/microorganisms10112217).
  • Please authorize each plant name in the WOF plant list (https://wfoplantlist.org/ )

1) In Table 1, the plant name is written half ‘Medicago’ Please check your cited article it mentions there ‘Medicago lupulina’…. The right name is Medicago lupulina L.

2) Table 2; The plant name mentioned is Pinus densiflora, however, the correct name is Pinus densiflora Siebold & Zucc. (https://wfoplantlist.org/taxon/wfo-0000481256-2024-06?page=1 )

Please check every name throughout the manuscript and correct it accordingly. very important

·       Table 2, column (heavy metals); names of heavy metals are not uniform i.e. Cd remediation, Cadmium (Cd), copper (Cu), Cu, Pb, Co, Cr, Fe, Mn and Ni, Copper ….. please follow a specific pattern i.e. Cd, Cu, Pb and/or  Cadmium, Copper, Lead………do the said throughout the manuscript….very important

·       This is wrong Table 2 ; (Zea mays L, please write Zea mays L., similarly, Trifolium pratense L.), it should be Trifolium pratense L.

·       These names should be authorized, accordingly. Nymphaea nymphaea, Bornmuellera tymphaea, Pteris vittate, Sedum alfredii and Thlaspi caerulescens (https://wfoplantlist.org/ ).

Thanks

Author Response

DEAR REVIEWER, THANK YOU AGAIN FOR YOUR COMMENTS, WE ARE ANSWERING AS FOLLOWS:

Dear Authors/Editor

Many thanks for considering my queries and kind responses. The article is undoubtedly much better now, however, still a few important corrections I think are mandatory to address.

  • What is the source of the figure, please mention,

RESPONSE: References included, thank you!

  • Previous comment(Some scientific names lacking authority names, more specifically in tables, please consider,

RESPONSE: Could you be so kind as to indicate which ones we should include? This could be very relative, but we tried to include good reputed journals instead of looking at the names, because we do not know them.)

  • As per your/authors' request, I am trying to explain this issue, please check table No. 1, how the plant names are mentioned (DOI: 10.3390/microorganisms10112217).
  • Please authorize each plant name in the WOF plant list (https://wfoplantlist.org/ )

1) In Table 1, the plant name is written half ‘Medicago’ Please check your cited article it mentions there ‘Medicago lupulina’…. The right name is Medicago lupulina L.

2) Table 2; The plant name mentioned is Pinus densiflora, however, the correct name is Pinus densiflora Siebold & Zucc. (https://wfoplantlist.org/taxon/wfo-0000481256-2024-06?page=1 )

Response: We apologize for the misunderstanding in your comment. The names have been corrected accordingly. Thank you very much for suggesting the taxonomic plant list webpage; we learned something new!

Please check every name throughout the manuscript and correct it accordingly. very important

  • Table 2, column (heavy metals); names of heavy metals are not uniform i.e. Cd remediation, Cadmium (Cd), copper (Cu), Cu, Pb, Co, Cr, Fe, Mn and Ni, Copper ….. please follow a specific pattern i.e. Cd, Cu, Pb and/orCadmium, Copper, Lead………do the said throughout the manuscript….very important
  • This is wrong Table 2 ; (Zea mays L, please write Zea maysL., similarly, Trifolium pratense L.), it should be Trifolium pratense L.
  • These names should be authorized, accordingly. Nymphaea nymphaea, Bornmuellera tymphaea, Pteris vittate, Sedum alfredii and Thlaspi caerulescens(https://wfoplantlist.org/ ).

Response: The names have been corrected accordingly. Thank you so much for your suggestion. 

Thanks

Reviewer 3 Report

Comments and Suggestions for Authors

The authors have improved the manuscript in the revised version, but some fragments still require corrections or comments (see below).

Remarks:

1. The authors did not understand my remark - lines 126 and 129: it should be written "tymphaea" instead of "nymphaea". As for the spelling of plant names, I find it incorrect to indicate full names for some plants (lines 125, 172, 180 and Table 2), and for some - names without authors.

2. The text fragment about Azospirillum spp. has been corrected, but not correctly. Since 2020, strains Sp7 and Sp245 have been divided into different species. Now, both strains are type strains for their species: Azospirillum brasilense Sp7T and Azospirillum baldaniorum Sp245T. Therefore, it is incorrect to call both strains Azospirillum brasilense (first version of the manuscript) or Azospirillum baldaniorum (revised version).

3. Table 3 for reference 126 lists "Arabidopsis thaliana and Nicotiana tabacum", but these are plants into which the merC gene from the bacterium Acidithiobacillus ferrooxidans was transformed. Accordingly, it is more correct to list the bacterium as the source of the gene.

4. Lines 74-75: References 11 and 12 do not list all the plants listed here.

5. The manuscript does not discuss the importance of genomic data for the early described heavy metal-resistant PGPR strains. I believe that the use of such data (along with metagenomic data for new isolates) can also contribute to the development of phytoremediation.

Author Response

DEAR REVIEWER, THANK YOU AGAIN FOR YOUR COMMENTS, WE ARE ASNWERING AS FOLLOWS:

REVIEWER 3

The authors have improved the manuscript in the revised version, but some fragments still require corrections or comments (see below).

Remarks:

  1. The authors did not understand my remark - lines 126 and 129: it should be written "tymphaea" instead of "nymphaea". As for the spelling of plant names, I find it incorrect to indicate full names for some plants (lines 125, 172, 180 and Table 2), and for some - names without authors.

RESPONSE: Response: We apologize for the misunderstanding in your comment. The names have been corrected accordingly, including Tables 1 and 2.

  1. The text fragment about Azospirillum spp. has been corrected, but not correctly. Since 2020, strains Sp7 and Sp245 have been divided into different species. Now, both strains are type strains for their species: Azospirillum brasilenseSp7T and Azospirillum baldaniorumSp245T. Therefore, it is incorrect to call both strains Azospirillum brasilense (first version of the manuscript) or Azospirillum baldaniorum (revised version).

RESPONSE: Thank you so much for your clarification, we were confused about it. We corrected the paragraph as follows: ¨The diazotrophic rhizobacteria Azospirillum baldaniorum Sp245 (formerly A. brasilense) and Azospirillum brasilense Sp7 are two species capable of colonizing different habitats. The former (Sp245) is considered a facultative endophyte, as it can live either as an endophyte or saprophyte, while the Sp7 strain can only colonize the surface of roots. In a study by Kamnev and colleagues [35], they observed that these Azospirillum strains respond differently to the presence of heavy metals. The wild-type Sp245 strain showed a less marked buildup of stress-induced compounds compared to the non-endophytic type strain Sp7 when exposed to metals such as Co²⁺, Cu²⁺, and Zn²⁺. Moreover, both strains revealed reduced indole-3-acetic acid (IAA) production in the presence of Cu²⁺ or Cd²⁺, which hypothetically affects their efficiency as plant growth promoters in soils contaminated with heavy metals.¨

  1. Table 3 for reference 126 lists "Arabidopsis thaliana and Nicotiana tabacum", but these are plants into which the merC gene from the bacterium Acidithiobacillus ferrooxidans was transformed. Accordingly, it is more correct to list the bacterium as the source of the gene.

RESPONSE: Modified as suggested, thanks!

  1. Lines 74-75: References 11 and 12 do not list all the plants listed here.

REPONSE: Thank you, more references were added.

  1. The manuscript does not discuss the importance of genomic data for the early described heavy metal-resistant PGPR strains. I believe that the use of such data (along with metagenomic data for new isolates) can also contribute to the development of phytoremediation.

RESPONSE: Yes, we agree with you. However, this topic is out of the scope of our work. We only mention the importance of such genomic tools in subheading 6. Assessment of perspectives, but thank you for your comment.
